Multisatellite observations of the magnetosphere response to changes in the solar wind and inter-
2                                              planetary magnetic field

Galina Korotova[1,2], David Sibeck[3], Scott Thaller[4] John Wygant[4], Harlan Spence[5], Craig Kletz-
ing[6], Vassilis Angelopoulos[7], and Robert Redmon[8]
[1]IPST, University of Maryland, College Park, MD, USA
[2]IZMIRAN, Russian Academy of Sciences, Moscow, Troitsk, Russia
[3]Code 674, NASA/GSFC, Greenbelt, MD, USA
[4]College of Science and Engineering, University of Minnesota, Minneapolis, MN, USA
[5]EOS, University of New Hampshire, Durham, NH, USA
[6]Department of Physics and Astronomy, Iowa University, Iowa City, IA, USA
[7]Department of Earth, Planetary and Space sciences, UCLA, Los Angeles, CA, USA
[8]Solar and Terrestrial Physics division, NGDC/NOAA, Boulder, CO, USA
**Abstract**
We employ multipoint observations of the Van Allen Probes, THEMIS, GOES and Cluster to
present case and statistical studies of the electromagnetic field, plasma and particle response to
interplanetary (IP) shocks observed by Wind. On February 27, 2014 the initial encounter of an
IP shock with the magnetopause occurred on the postnoon magnetosphere, consistent with the
observed alignment of the shock with the spiral IMF. The dayside equatorial magnetosphere
exhibited a dusk-dawn oscillatory electrical field with a period of ~ 330 s and peak to peak
amplitudes of ~ 15 mV/m for a period of 30 min. The intensity of electrons in the energy
range from 31.5 to 342 KeV responded with periods corresponding to the shock induced ULF
electric field waves. We then perform a statistical study of Ey variations of the electric field
and associated plasma drift flow velocities for 60 magnetospheric events during the passage of
interplanetary shocks. The Ey perturbations are negative (dusk-to-dawn) in the dayside
magnetosphere (followed by positive or oscillatory perturbations) and dominately positive
(dawn-to-dusk direction) in the nightside magnetosphere, particularly near the Sun-Earth line
within an L-shell range from 2.5 to 5. The typical observed amplitudes range from 0.2 to 6
mV/m but can reach 12 mV during strong magnetic storms. We show that electric field
perturbations increase with solar wind pressure and that the changes are especially marked in
the dayside magnetosphere. The direction of the Vx component of plasma flow is in agreement
with the direction of the Ey component and is antisunward at all local times except the nightside

magnetosphere, where it is sunward near the Sun-Earth line. The flow velocities Vx   range from 0. 2 to 40 km/s and are a factor of  5 to 10 times stronger near noon as they correspond to  greater variations  of the  electric field  in this region. We demonstrate that the shock-induced electric field signatures  can be classified into four different groups according to the initial Ey electric field response and these  signatures are local time dependent.  Negative and bipolar pulses predominate on the dayside with positive pulses occur on the nightside.  The  ULF electric field pulsations  of Pc and Pi types  produced by IP shocks are observed at all local times and  in the range of  periods  from several tens of seconds to several minutes. We believe that that  most electric field pulsations  of the Pc5 type in the dayside magnetosphere  at L < 6  are produced by field line resonances.  We show that the direction of the shock normal determines the direction of the propagation of the shock-induced magnetic and plasma disturbances.  The observed direc-tions of velocity Vy predominately agree with those expected for the given spiral or orthospiral shock normal orientation.

**Keywords:** Interplanetary shocks, solar wind – magnetosphere interactions, energetic particles, trapped.

**1 Introduction**

Sudden increases in the solar wind dynamic pressure accompanying   interplanetary (IP) shocks cause earthward motion of the bow shock and the magnetopause  and launch fast and in-termediate mode waves into the magnetosphere  (Tamao, 1964). The fast mode  waves propagate both radially inward and azimuthally around the Earth (Araki et al., 1997) whereas the interme-diate mode waves propagate along magnetic field lines to produce transient perturbations in the high-latitude dayside ionosphere (Southwood and Kivelson, 1990; Glaßmeier and Heppner, 1992).  Using multipoint observations ) estimated the propagation speeds to be about 600 km/s in the radial direction from geostationary orbit to the ground and about 910 km/s in the  azimuthal direction in the equatorial plane.  Nopper et al. (1982) estimated an impulse disturbance speed of about 1500 km/s at geostationary orbit.  Schmidt and Pedersen (1988) derived a propagation ve-locity for the radially inward travelling compressive wave of 950 km/s and for  the azimuthal wave in the outer magnetosphere  of 1100 km/s. Samsonov et al. (2007) used a magnetohydro-

dynamic code to simulate the interaction of a moderately strong interplanetary shock propagat-
ing along the Sun-Earth line and obtained the average speed of the primary and reflected fast
shocks in the magnetosphere to be about 700 km/s, in agreement with their assumptions con-
cerning the mean Alfvén velocity in the outer dayside magnetosphere (1000 km/s) and in the
plasmasphere (500 km/s).
The IP shock orientation plays an important role in determining the associated geophysical
effects (e.g., Oliveira and Raeder, 2015) showed that system evolution times are much longer for
shocks with normals oblique to the Sun–Earth line. The pressure pulse model of Sibeck (1990)
predicts dawnward moving transient events near local noon when shock normals point per-
pendicular to the nominal spiral interplanetary magnetic field (IMF) direction, but duskward
moving events occur near local noon for events when shock normals point perpendicular to the
orthospiral IMF orientation. The direction of the plasma flow within the magnetosphere is ex-
pected to be consistent with the orientation of the shock. That is to say dawnward flow for spi-
ral IMF shocks and duskward flow for orthospiral IMF shocks. Here orthospiral refers to IMF
longitudes ($0º < \Lambda < 90º$ and $180º < \Lambda < 270º$), spiral refers to IMF longitudes ($90º < \Lambda < 180º$
and $270º < \Lambda < 360º$), where longitude $\Lambda = 0º$ points sunward, and $\Lambda = 90º$ duskward.
The magnetic and electric fields are key parameters for understanding of the response of
the Earth's space environment to IP shocks. The propagation and evolution of electric fields in
the magnetosphere-ionosphere system in response to IP shocks have been studied for several
decades but signatures of the shock related electric field perturbations are still not fully under-
stood. Knott et al. (1985) reported that the electric field observed by the GEOS-2 satellite
showed a transient signature of about 7 mV/m in the dayside magnetosphere associated with the
onset of a Sudden Commencement (SC). These signatures were followed by Pc4-5 oscillations.
Schmidt and Pedersen (1988) performed a statistical investigation of the GEOS2 electric field
signatures associated with SC that showed a clear tailward flow pattern near local noon. Close to
the flanks or in the nightside of the magnetosphere the corresponding flows also exhibited a
radially inward component. Shinbory et al (2004) investigated the detailed signatures of the
Akebono electric and magnetic fields associated with SCs inside the plasmasphere ($L < 5$). The
initial excursion of the electric field associated with SCs was almost directed westward at all lo-
cal times. The amplitude did not show a clear dependence on magnetic local time and the inten-
sity of the Ey field gradually increased by 0.5-2.0 mV/m about 1-2 minutes after the onset of the
initial electric field impulse.  The propagation velocity of SCs disturbances derived from the am-
plitude ratio of the electric field to magnetic field was about 360 km/s in the equatorial  plasmas-
phere.  Kim et al. (2009) used  an MHD simulation to examine the electric field and suggested
that the SC associated electric field seen by Shinbory et al. (2004) was the convection electric
field. Takahashi et al. (2017) investigated the spatial and temporal evolution of large-scale elec-
tric fields in the magnetosphere and ionosphere associated with SCs using multipoint equatorial
magnetospheric and ionospheric satellites together with  ground radars and showed that the
propagation characteristics of electric fields in the equatorial plane depend on magnetic local
time.  They showed  that the initial variation of the electric field  (negative Ey) lasted  about one
minute  and was directed westward  throughout the inner magnetosphere.  Positive Ey became
dominant 2 min after SCs  propagated  to pre-midnight or post-midnight region with near costant
amplitude.
Observations and MHD simulations (e.g., Li et al., 1993;  Zong et al., 2009; Halford et al.,
2014; Schiller et al., 2016) show that  the electric fields  generated  by  sudden compressions can
resonantly interact with trapped charged particle populations within the Earth magnetosphere,
energizing and    injecting  them deep into the magnetosphere. During the well-known shock
event in March 1991, the CRESS satellite observed injected electrons energized to extremely
high energies, up to 5  MeV (Blake et al., 1992). Wygant et al. (1994) showed that  the shock
related   electric and magnetic field perturbations observed by the CRRES satellite  in the
nightside inner magnetosphere exhibited a bipolar  waveform with amplitude of about 80 mV/m
and 140 nT, respectively, and energized the energetic electrons to energies  up to 15 MeV. Foster
et al.  (2015) found that a shock with  an  azimuthal electric field impulse of 10 mV/m  observed
by the Van Allen Probes  was responsible for  accelerating  1.5-4.5 MeV electrons by 400 KeV
in the radial region of  L= 3.5-4.
This paper focuses on two major issues. We will inspect multispacecraft  electric and mag-
netic  field and  particles and  plasma  observations to study their response to an IP shock on Feb-
ruary 27, 2014. We will time the occurrence of magnetic field disturbances associated with the
shock in space and the magnetosphere and will show that it  propagated dawnward  consistent
with expectations based on the shock orientation. Then we will perform a statistical study of the
Van Allen Probes electric field disturbances in the magnetosphere and associated plasma drift
Vx and Vy velocities in response to IP shocks. We will show that there are four categories of
electric field perturbations that occur in response to shock-induced compressions and that these
signatures have a clear dependence on magnetic local time. We will show that the direction of
the shock normal has an important effect on the propagation of the shock induced magnetic and
plasma disturbances and that our statistical results are consistent with MHD simulation predic-
tion.

## 133 2 Data sets

The extensive Van Allen Probes, THEMIS, Cluster and GOES multi-instrument data sets
provide numerous opportunities to observe the magnetospheric response to the changes in the
solar wind and interplanetary magnetic field monitored by Wind. The five THEMIS spacecraft
were launched in 2007 and carry identical instruments and operated in highly elliptical, near-
equatorial, orbits that precess about the Earth with apogees of 12, 20, and 30 Re and orbital peri-
ods of 1, 2, and 4 days. With the outermost two spacecraft ARTEMIS now at the Moon, three
THEMIS spacecraft remain on the innermost orbits. We use magnetic field data with 3 s time
resolution from the THEMIS FGM triaxial fluxgate magnetometers (Auster et al., 2008). The
ESA electrostatic analyzer on the THEMIS spacecraft measures the distribution functions of
0.005 to 25 keV ions and 0.005 to 30 keV electrons over 4Pi-str and provides accurate 3 s time
resolution plasma moments, pitch angle and gyrophase particle distributions (McFadden et al.,

145 2008).

The two Van Allen Probes were launched in August 2012 into nearly identical equatorial
and low inclination (~10°) orbits with perigee altitudes of 605 and 625 km and apogee altitudes
of 30410 and 30540 km (Mauk et al., 2012). Both satellites carry identical sets of instruments to
measure charged particle populations, fields, and waves in the inner magnetosphere. In this pa-
per, we employ observations from the Energetic Particle, Composition, and Thermal Plasma
Suite (ECT: MagEIS, 20-4000 keV for electrons) (Spence et al., 2013; Blake et al., 2013). Elec-
tric and Magnetic Field Instrument Suite and Integrated Science (EMFISIS) (Kletzing et al.,
2013), and the Electric Field and Waves Suite (EFW) (Wygant et al., 2013). In particular, we
inspect electric and magnetic field observations with 11 and 4 s time resolution, respectively,

and differential particle flux measurements with ~ 11 s (spin period) time resolution. The electric field data were obtained from sites http://www.space.umn.edu/rbspefw-data and CDAWEB where they are presented in an MGSE (modified GSE) coordinate system. They provide two components Y and Z of the electric field. Both components are in the spin plane of the spaceraft and are measured with the 50 m long booms. The spin axis X is oriented within 37 degrees of the Earth-Sun line. The spin axis component of the electric field can be obtained from the E dot B = 0 assumption. For this to succeed the magnetic field should be at least 15 degrees out of the spin plane. To calculate Van Allen Probes plasma flow velocities we converted the electric field data from modified MGSE coordinates into GSE coordinates. Additionally we used magnetic field data from GOES 13 and 15 with 0.5 s time resolution (Singer et al., 1966) and Cluster with 4 s time resolution (Balogh et al., 1997). We use Wind solar wind magnetic field and SWE plasma data with 3 s (Lepping et al., 1995) and 1 min, respectively (Ogilvie et al., 1995).

**3 Observations**

Figure 1 presents Wind magnetic and plasma data from 15:30 to 16:10 UT on February 27, 2014. The arrival of the shock at Wind at 15:50 UT (X, Y, Z GSM = (220.9, 93.9, 30.7 Re)) is revealed by an enhancement in the interplanetary magnetic field strength from 6 to 16 nT and total plasma velocity from 350 to 420 km/s. The IMF had positive Bx and negative By components during the whole interval that both increased the shock arrived. The solar wind density increased from 18 to 45 cm-3, and the dynamic pressure increased from 3 to 13 nPa. This fast forward (FF) shock was oblique. Its normal was calculated using magnetic field coplanarity and pointed in the GSM [nx, ny, nz] =[-0.8, -0.4, -0.3] direction, i.e., antisunward, dawnward, and southward. Consequently the shock should first strike the northern dusk bow shock and magnetopause. i.e., it has a spiral IMF orientation. We will use the direction of the shock normal to interpret the timing results for the IP shock arrival observed by THEMIS, GOES, Cluster and the Van Allen Probes spacecraft for this event.

Figure 2 shows the GSM locations of The THEMIS, Cluster, Van Allen Probes and GOES spacecraft at ~ 16:50 UT (Their coordinates are given in Table 1). All the spacecraft located in the solar wind observed the enhanced magnetic field strength, densities, velocities and temperatures associated with the IP shock. The shock induced disturbances were seen just upstream

from the bow shock by Cluster 1 and 3, located at high southern postnoon latitudes at 16:48:46
UT and 16:48:57 UT, respectively .
Figures 3 (a, b) show the THEMIS D and A observations of the magnetic field, plasma and
energy spectra of ion fluxes from 16:40 to 17:20 UT. The spacecraft were initially located in the
magnetosheath. At 16:49:04 UT the IP shock hit THEMIS D as indicated by enhanced densities,
magnetic field strength and velocities. Particles from low to high energies showed the increase
of energy and enhanced fluxes. The shock produced compression caused the bow shock to move
inward at 16:49:36 UT, past the spacecraft as indicated by the decrease in the magnetic field
strength and, decrease in density and temperature and spectra expected for its entry into the solar
wind. THEMIS A observed the IP shock at 16:49:12 UT and in about 1 min and 34 s later its
magnetic field, density and temperature traces indicate that the bow shock moved inward past
THEMIS A.
Figures 4 (a, b) show GOES 13 and 15 observations of the magnetic field from 16:40 to
17:20 UT. Following the arrival of the transmitted IP shock at GOES 13 near local noon at
16:50:07 UT there was a sharp increase of magnetic field variations with amplitudes of ~70 nT
in the H component. The shock induced compression was so strong that at 17:02 UT GOES 13
briefly entered the sheath. The shock front was then detected at GOES 15 in the morning local
hours 33 sec later at 16:50:40 UT, where it caused a gradual increase of the magnetic field am-
plitudes by ~ 20 nT followed by compressional pulsations that fall in the category of Pc5 pulsa-
tions.
The upper and middle panels of Fig. 5 (a, b) present the Van Allen Probes A and B mag-
netic field and electric observations from 16:40 to 17:20 UT. The arrival of the shock character-
ized by a strong (~ 50 nT) increase in the total magnetic field strength and bipolar variations in
all three components of the electric field at ~ 16:50:26 UT at Probe B and 7 sec later at Probe
A. The initial electric field perturbations in the $E_y$ component observed by Van Allen Probes A
and B were directed dawnward with amplitudes of -9.4 and - 8.2 mV/m, respectively, but ~ 4
minutes later the sense changed direction towards dusk (with amplitudes of 5.3 and 5.8 mV/m).
We interpret these variations as due to a compression of the magnetosphere followed by a reflec-
tion (Samsonov et al., 2007). The $E_z$ and $E_x$ components show variations with amplitudes that
are a factor of 1.5-2 smaller than those of the $E_y$ component. The bipolar electric field wave-

forms are   followed by geomagnetic pulsations  with periods of  ~ 330 s   that damp within ~ 30 min.

Figures 6 (a, b) present Van Allen Probes A and B observations of the Ez component of the electric field and pitch angle distributions for electron energies of 31.5, 53.8, 108.3, 183.4, 231.8, and 342 KeV measured by the MagEIS instrument.  The electrons exhibit enhanced intensities at all energies but the  most intense occur at pitch angles near $90^{o}$, immediately after the arrival of the IP shock. Kanekal et al. (2016) suggested that the shock-injection mechanism can be effective for energizing particles over a substantial range of pitch angles. The initial flux enhancement is more pronounced by comparison with the following pulses.   One of the interesting feature in Figures 6 (a, b) is that the  intensity of  electrons  in the energy range of  31.5-342 KeV  exhibits a regular periodicity  with periods  corresponding to  the ULF electric field  waves.  The oscillations in electron fluxes are in quadrature with the Ey component.   This component is of special interest because some charged particles that drift azimuthally as a consequence of the gradient and curvature drifts in the Earth magnetic field can traverse this electric field acquiring  a significant amount of energy. Figures 7 (a, b, c, d, e, f) present the response of the energetic electrons to the IP shock in the energy range from 31 to 183 keV.  Panels d and b show that after the shock arrival the electron population increased, especially  for the lower energies.  In the electrical field of 15 mV/m  electron fluxes increased by factors of 21 and 14 at Van Allen  Probes B and A, respectively, in less than a drift period (panels e and f).  The energetic electron fluxes do not display obvious phase differences across the energies. We interpret these observations as evidence for prompt energization of electrons due to shock induced ULF electric fields with an additional contribution for the initial acceleration from the compressional effect of the shock. It should be noted that electrons can be accelerated most significantly via drift resonance (Southwood and Kivelson, 1981) when resonant particles drift with the same velocity as the wave front. Claudepierre et al. (2013) showed Van Allen Probes observations of the energy dependence of the amplitude and phase of the electron flux modulations which were consequences of drift resonance between ~ 60 keV electrons and fundamental poloidal Pc5 waves. Hao et al. (2014) presented Van Allen Probes observations of electron injections caused by the IP shock and showed that the injected electrons with energies between 150 KeV and 230 KeV were in drift resonance with the excited poloidal ULF waves.  Considering the process for energizing drift resonant electrons, the

value for the E x B drift velocities of the particles in the wave fields provides important information. We calculated the Vx and Vy drift velocities at Van Allen Probes A and B for the interval from 16:40 to 17:20 UT and present them in the two bottom panels of Fig. 5 (a, b). The Vx and Vy components associated with the minimum peak of the Ey electric field are about -40 km/s and -15 km/s for Van Allen Probe B and -35 km/s and -6 km/s for Van Allen Probe A, i. e., the initial direction of the plasma flow is tailward and dawnward consistent with expectation for the spiral orientation of the IP shock.

Interaction with the initial fast mode pulse and subsequent ULF electrical field oscillations can have an important effect on particle acceleration. In considering the energization of electrons on February 27, 2014, an encounter with the observed electric field for a period of 240 s will transport the electrons earthward by $\delta Re = 1.3$ to 1.6 Re from their original position at L = 6.4 for Van Allen Probe A and at L = 7.1 for Van Allen Probe B. Conservation of the first adiabatic invariant implies that such particles will be energized by a factor of about 1.9 - 2.3 in only one cycle of the electric field pulsations. The studies of Wygant et al. (1994) using CRRES data and Foster et al. (2015) using Van Allen Probes data, and others have demonstrated that the tailward propagation of the strong shock-induced electric field impulse and subsequent ULF processes can result in the extremely fast acceleration of relativistic electron populations inside the plasmasphere.

Knowing the distances between the satellites and the lag times for the propagation of shock induced disturbances we calculated the shock propagation velocities. Table 1 summarizes the onset times of the shock driven encounters at different spacecraft. In the solar wind Cluster 1 observed the shock earlier than Cluster 3, respectively, that is the shock moved dawnward. The shock perturbations occurred almost simultaneously in the magnetosheath at Themis A and D (delta t < 10 s) suggesting the front strikes a broad region of the magnetopause at once. The shock induced impulse propagated antisunward, southward and both dawnward and, presumably, duskward (thick arrows in Fig. 2) from the point of origin on the magnetopause (depicted as a red oval in Fig. 2) that is consistent with the orientation of the IP shock. In the outer magnetosphere the propagation velocity for the disturbance was about 1348 km/s between Goes 13 and 15 but only about 390 km/s between Van Allen Probes B and A. We believe that the shock induced pulse propagated with the velocity of fast mode waves. The local fast mode speed

can be evaluated from Van Allen Probe measurements of the magnetic field and density. At the
time of the shock encounter Van Allen Probes A and B were in the high-density plasmasphere at
L = 5.5 and L = 5.1, respectively. For a measured local magnetic field of 255 nT for Probe A
and 220 nT for Probe B and density of ~ 200 cm-3 derived from the potential of both space-
craft, the fast-mode speeds will be ~ 395 km/s and 337 km/s, respectively, which are con-
sistent with our estimates of the propagation velocity derived from the time difference of shock
arrivals at the spacecraft. The decrease of the fast mode wave speed in the plasmasphere relative
to that in the outer magnetosphere agrees well with earlier studies (e.g., Wilken, 1982; Foster et
al., 2016).

**4 Statistical study of shock-initiated signatures of the electric field**
The list of IP shocks used in this study was obtained from Heliospheric Shock Database
mantained and generated by the University of Helsinki [http://ipshocks.fi]. They identify shocks
by visual inspection and an automated shock detection algorithm. To be included in the data-
base a shock should satisfy the following upstream to downstream jump conditions: Bdown/Bup
> 1.2, Ndown/Nup > 1.2, Tdown/ Tup > 1/1.2, for FF Vup-Vdown >20 km/s. The normal vec-
tor of the shock (n) was calculated from the magnetic field data and velocities using the mixed
mode method (Abraham-Shrauner and Yun, 1976). When there is data gap in the velocity com-
ponents the normal was calculated using magnetic field coplanarity (Colburn and Sonett, 1966).
In view of the importance of the electric field in energizing particles we performed a statis-
tical study of Ey variations of the electric field and associated plasma drift Vx and Vy velocities
during the passage of interplanetary shocks We identified more than 60 events observed by Van
Allen Probes A and B associated with FF IP shocks for the period from 2013 to 2015. The
shocks arrived from Wind with lag times in the time range from 26 min to 58 min and pro-
duced magnetic field perturbations in the magnetosphere from several to 130 nT. Discontinui-
ties in the solar wind plasma such as shocks have often been considered as possible triggers for
the release of energy stored within the magnetotail in the form of magnetospheric substorms.
Most previous studies of shocks leading to substorms have relied on ground magnetometer ob-
servations. Recently it has been shown that the use of global auroral images to identify substorm
onsets has some advantages over many other alternative substorm onset signatures, such as low-

latitude Pi2 pulsations, auroral kilometric radiation (AKR), and dispersionless particle injections and magnetic field dipolarization at geosynchronous orbits (e.g., Liou et al., 2000). To identify substorms triggered by shocks in our study we considered negative magnetic bays by examining the westward auroral electrojet AL index at the times when SSC were determined from low-latitude ground magnetograms.. As a quantitative definition for the substorm bay does not exist we used the criteria of Liou et el. (2003) that there should be a sharp decrease in AL of at least 100 nT occurring within a 20 min window starting at the SSC. We found that shocks triggered a substorm in the magnetosphere in 17 of the 30 examined events. Further study whether these negative magnetic bays are unique identifiers of substorms is beyond the scope of the paper. Other effects in the magnetosphere initiated by IP shocks are perturbations in the electric field (Wygant et al., 1994) and the radiation belt (Blake et al., 2015). Understanding and predicting such responses is important for reducing the risks associated with space exploration. We found that 55 events showed an electron enhancement at energies of 32-54 keV measured by MagEIS at all local time and three of them were accompanied by intensity decreases at higher energies. Five events showed a decrease of the 32-54 keV energy electrons observed in the nightside magnetosphere.

The passage of a shock causes electric field perturbations and their amplitudes to increase in proportion to the intensity of the IP shocks. The E field vectors prior to each compression differ greatly from those during the compressional activity. We classified the shock-induced electric field signatures into four different groups according to the examples presented in the upper panels of Fig. 8. Group A presents a negative pulse in the Ey component, B group presents a negative-positive pulse, C group presents a positive pulse and D group presents pulsations. Figure 9 presents occurrence patterns for events with the four different signatures of the electric field initiated by IP shocks. It provides evidence that they are local time dependent. Negative and bipolar pulses predominate on the dayside with positive pulses occur on the nightside. The ULF electric field pulsations of Pc and Pi types produced by IP shocks are observed at all local times and in the range of periods from several tens of seconds to several minutes. We believe that the magnetic field as well the electric field pulsations initiated by IP shocks are generated by a wide variety of mechanisms including plasma instabilities, transient reconnection, pressure pulses, and often correspond to field line resonances. Their characteris-

tic features are determined to large extent by local time. In the dayside magnetosphere typi-
cal pulsations are of the Pc5 type. Sometimes they last for more than twenty wave cycles
without noticeable damping which could be explained by a continuous input of the solar wind
energy into the magnetosphere. In the nightside magnetosphere during substorms, the generation
of Pi2 pulsations is more common. They exhibit an irregular form, last 3-5 wave cycles, and
often exhibit damping. Figure 10 presents periods of the pulsations ( measured for the first wave
cycle of oscillations) as a function of radius and shows that periods increase with increasing
radius. A simple explanation for this behavior of pulsation frequencies with radial distance can
be given in terms of standing Alfvén waves along resonant field lines (Sugiura and Wilson,
1964). The length of the field lines, the magnetic field strength, and the plasma density distribu-
tion determine the Alfvén velocity, and the periods of the pulsations. This plot indicates that
most electric field pulsations of the Pc5 type in the dayside magnetosphere at L < 6 are pro-
duced by field line resonances.
Figure 11 presents the amplitudes and direction of the initial Ey response to IP shocks in
the X-Y GSM plane. The perturbations are negative (dusk-to-dawn) in the dayside
magnetosphere (followed by the positive or oscillatory perturbations) and dominately positive
(dawn-to-dusk direction) in the nightside magnetosphere, particularly mostly near the Sun-
Earth line within an L-shell range from 2.5 to 5. The typical observed amplitudes range from
0.2 to 6 mV/m but can reach 12 mV during strong magnetic storms. In the nightside
magnetosphere the response of Ey is rather weak and its ampludues do not not exceed 3 mV/m.
To demonstrate the impact of IP shocks Fig. 12 shows amplitudes of the initial electric field
variations (blue and red crosses) as a function of dynamic pressue observed at Wind. The
electric field perturbations increase with the solar wind pressure and that the changes are
especially marked in the dayside magnetosphere (red points) as this region is more fully
exposed to compression than the nightside sector that is shielded from the frontside
compression.
To determine the Vx direction of the plasma after the impact of IP shocks we used the
formula $V = E \times B/B^2$ for the 60 events under the study. Figure 13 presents the amplitudes
and direction of the plasma drift velocities Vx that occur in response to IP shocks (in red -
sunward and in blue – tailward directions). The direction of the Vx component of plasma flow
is in agreement with the direction of the Ey component (except three peculiar events) and is
antisunward at all local times except the nightside magnetosphere, where it is sunward near the
Sun-Earth line. The tailward velocities are associated with tailward magnetic field line motion
in the dayside magnetosphere. Numbers show that the magnitudes of the flow velocities Vx
range from 0.2 to 40 km/s and are a factor of 5 to 10 times stronger near noon as they
correspond to greater variations of the electric field in this region.
Our results are consistent with the results of global 3D MHD code simulation for the geo-
synchronous magnetic field response in the nightside magnetosphere to IP shocks by Wang et
al. (2010) presented in Fig. 14. The figure shows contours of delta IBzI and velocity vectors in
the equatorial plane (blue regions - Bz negative, red regions - Bz positive). Their model re-
vealed that when a IP shock sweeps over the magnetosphere there are mainly two regions in the
nightside magnetosphere, a positive response region in Bz caused by the compressive effect of
the shock and a negative response region (blue) which is associated with the temporary en-
hancement of earthward convection. They believe that the displacement of the nightside magne-
topause caused by the IP shock launches a flow in the magnetosphere near the magnetopause
that has a significant y-component, and converges toward the X axis. In the vicinity of the Sun-
Earth line at ~ -5, -6 Re the flow diverges, producing both an earthward flow (consistent with
the sunward direction of plasma flow in the nightside magnetosphere presented in Fig. 13) and
a tailward flows.
As the direction of the shock normal should determine the direction of propagation of
transient perturbations and expected flow direction in the magnetosphere initiated by an IP shock
we calculated plasma drift velocities Vy for 30 events for which the Ex component could be
obtained from E dot B = 0. We categorized them into two groups for spiral and orthospiral ori-
entation of the shock normal. Figure 15 presents the amplitudes and direction of the plasma drift
velocities Vy observed by Van Allen Probes A and B in response to IP shocks (red- sunward
Vx and blue – tailward Vx directions) for spiral and orthospiral orientations of IP shocks. We
excluded several events from the list of shocks that lacked well defined shock normal. As an-
ticipated, the shock orientation controls the sense of dawn/dusk flows in magnetosphere. The
observed directions of velocity Vy predominately agree with those expected for the given shock
normal orientation: dawnward for shocks that sweeps dawnward across the magnetosphere,
duskward for shocks that sweep duskward.
**5 Conclusions**
We presented multipoint observations concerning the response of the electric and magnetic
fields, plasma and particles in the magnetosphere to an IP shock on February 27, 2014. We used
a multi-spacecraft timing method to determine the propagation speed and direction of the wave
front induced by the IP shock. The propagation velocity of the disturbances was about 1348
km/s between Goes 13 and 15 in the outer magnetosphere, but it was only about 390 km/s be-
tween Van Allen Probes B and A in the inner magnetosphere consistent with expectations for a
plasmasphere with limited radial extent. We deduced that the initial encounter of the IP shock
with the magnetopause occurred on the post-noon magnetosphere and the shock induced im-
pulse propagated as a fast mode wave both dawnward and, presumamly, duskward from the
point of origin consistent with the spiral orientation of the IP shock. The multipoint measure-
ments provide evidence for a dusk-dawn oscillatory electrical field in the dayside equatorial
magnetosphere with a peak-to-peak amplitude of $\sim$ 15 mV/m for a period of 30 min. Both
spacecraft observed enhanced fluxes of energetic electrons in the range of energies from 31.5 -
342 KeV and their intensity shows a regular periodicity with periods corresponding to the elec-
tric field pulsations. We interpret these observations as evidence for prompt energization of elec-
trons due to shock induced ULF electric fields with an additional contribution for the initial ac-
celeration from the compressional effect of the shock. An encounter with the observed electric
field for a period of 240 s will transport the electrons earthward by $\delta Re = 1.3$ to 1.6 Re from their
original positions at L = 6.4 for Van Allen Probe A and at L = 7.1 at Van Allen B. Conservation
of the first adiabatic invariant implies that such a particle will be energized by a factor of about
1.9 - 2.3 in only one cycle of the electric field pulsations. The initial plasma flow velocity in the
magnetosphere was directed tailward and dawnward, consistent with expectation for the spiral
orientation of the IP shock.
We identified more than 60 events observed by Van Allen Probes A and B associated with
FF IP shocks for the period from 2013 to 2015. The shocks arrived from Wind with lag times
in the time range from 26 min to 58 min and produced magnetic field perturbations in the
magnetosphere from several to 130 nT. We found that shocks triggered a substorm in the mag-
netosphere in 17 of the 30 examined events.    Taking advantage of the multipoint Van Allen
Probes observations, we performed a statistical study of Ey variations of the electric field and
associated plasma drift Vx and Vy flow velocities  during the passage of interplanetary shocks.
The  Ey perturbations  are  negative (dusk-to-dawn)   in the dayside magnetosphere (followed by
positive or oscillatory perturbations) and   dominately positive (dawn-to-dusk direction) in the
nightside magnetosphere, particularly  near the Sun-Earth line  within  an L-shell range  from 2.5
to 5.  The  typical observed amplitudes  range from 0.2 to 6 mV/m but can reach  12 mV during
strong magnetic storms.  We showed that  electric field perturbations  increase  with solar wind
pressure   and that the changes     are especially marked in the   dayside magnetosphere. The
direction of  the Vx component of plasma flow  is in agreement with the direction of the Ey
component and is  antisunward at all local times except the nightside magnetosphere, where it is
sunward near the Sun-Earth line but antisunward towards dawn and dusk. The flow velocities Vx
range from 0. 2 to 40 km/s and are a factor of  5 to 10 times stronger near noon  as they
correspond to  greater variations  of the  electric field  in this region**.** We investigated how the
electric field perturbations deviate from the preceding undisturbed period and demonstrated that
the shock-induced electric field signatures  can be classified into four different groups according
to the initial Ey electric field response. These signatures are local time dependent. Negative and
bipolar pulses predominate on the dayside with positive pulses occur on the nightside.  The  ULF
electric field  pulsations  of Pc and Pi types  produced by IP shocks are observed at all local times
and  in the range of  periods  from several tens of seconds to several minutes. We believe that
that  most electric field pulsations  of the Pc5 type in the dayside magnetosphere  at L < 6  are
produced by  field line resonances. One of the most important results from the present study is
that the direction of the shock normal determines the direction of the propagation of the shock
induced magnetic and plasma disturbances.  The observed directions of velocity Vy predomi-
nately agree with those expected for the given spiral or orthospiral shock normal orientation.
Our results are consistent with the results of global  MHD code simulation of the geosynchro-
nous nightside magnetic field response to IP shock by Wang et al. (2010).
Table 1. Times of encounter of the IP shock with the spacecraft and their locations in GSM
coordinates

| S/C | Time | Position | GSM [X, Y, Z] Re | |
|---|---|---|---|---|
| Wind | 15:50:12 | 220.90 | 93.92 | 31.49 |
| Cluster 1 | 16:48:46 | 13.10 | 7.82 | -9.44 |
| Cluster 3 | 16:48:57 | 12.60 | 7.73 | -10.16 |
| THEMIS D | 16:49:04 | 11.03 | 0.48 | 1.10 |
| THEMIS A | 16:49:12 | 9.22 | 4.39 | 0.53 |
| Probe A | 16:50:33 | 4.86 | -1.69 | 0.12 |
| Probe B | 16:50:26 | 5.33 | -1.39 | -0.10 |
| GOES 13 | 16:50:07 | 6.51 | -0.60 | 0.99 |
| GOES 15 | 16:50:40 | 2.71 | -6.02 | 0.45 |


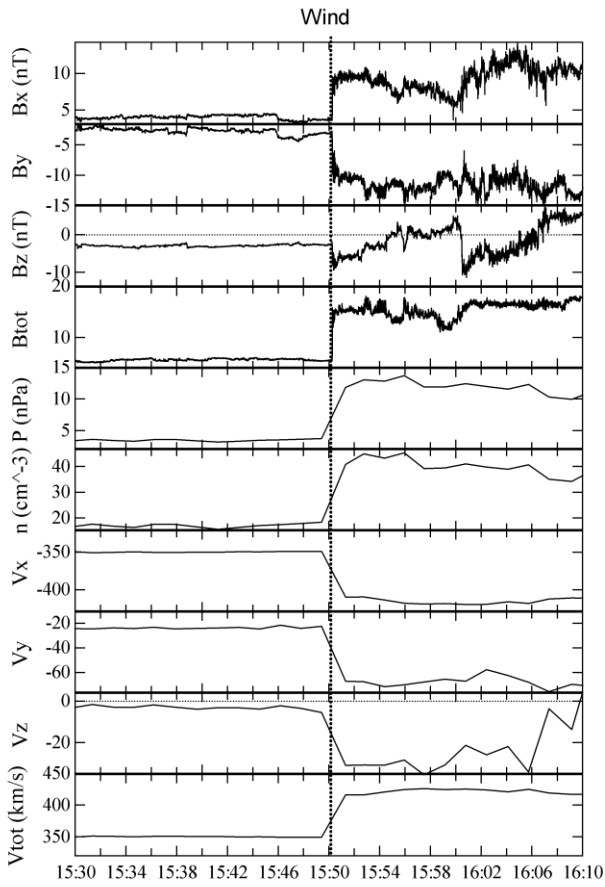


Figure 1. Wind observations of magnetic field and plasma in GSM coordinates from 15:30 UT
to 16:10 UT on February 27, 2014. Dashed line shows the time of arrival of an interplanetary
shock.




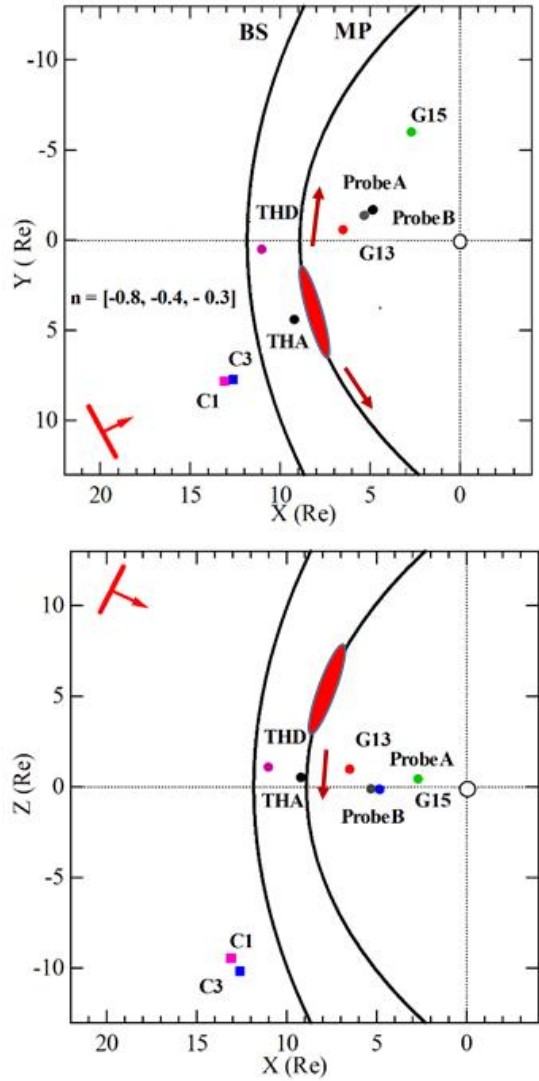


Figure 2. GSM locations of Cluster 1 and 3, THEMIS A, D, Van Allen Probes A and B and
GOES 13 and 15 in the X-Y and Z-Y GSM planes at ~ 1650 UT on February 27, 2014. The
meaning of the solid oval and thick arrows will be discussed in the text later.

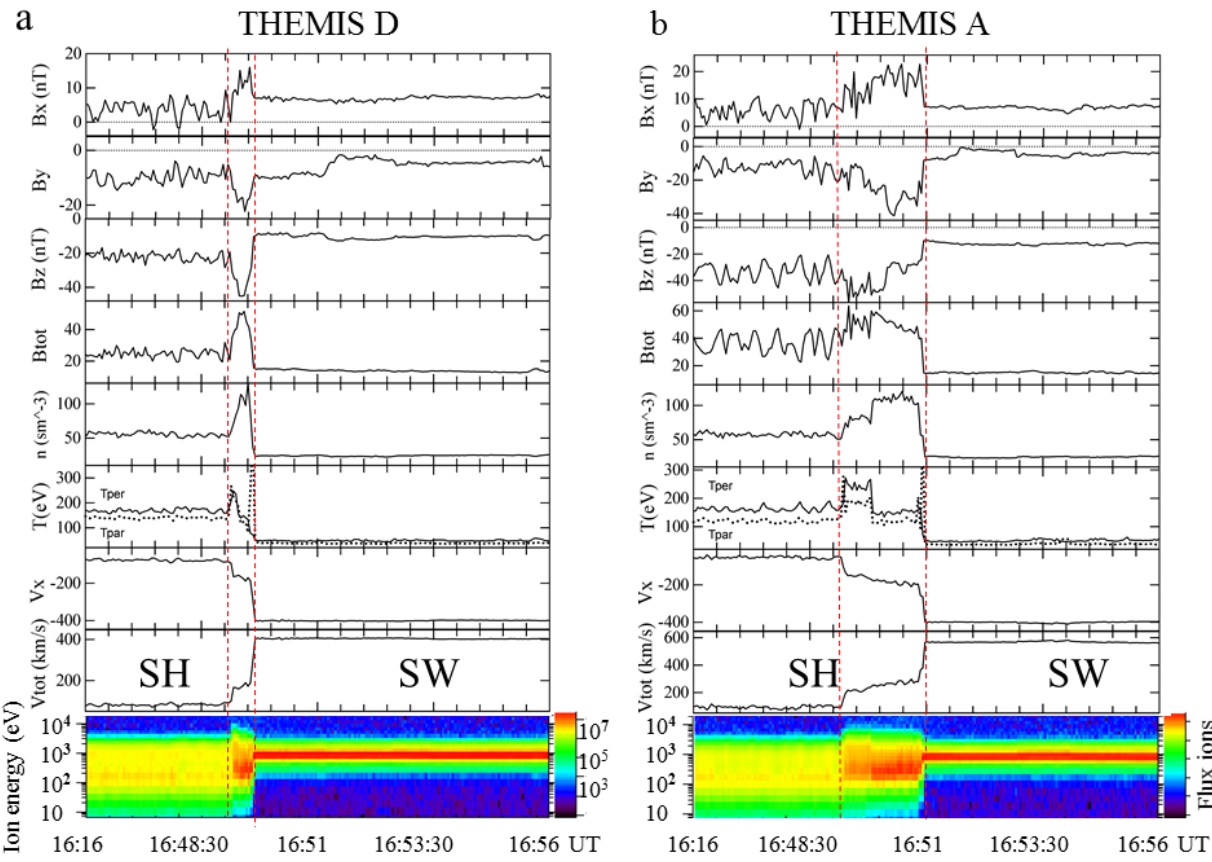

Figures 3 (a, b). THEMIS A (a) and THEMIS D (b) observations of magnetic field in GSM coordinates plasma and energy spectra of ion fluxes from 16:16 UT to 16:56 UT on February 27, 2014. At 16:49:01 UT the IP shock hit THEMIS D as indicated by enhanced densities, magnetic field strength and velocities. Particles from low to high energies showed the increase of energy and enhanced fluxes. The shock produced compression caused the bow shock to move inward at 16:49:36 UT, past the spacecraft as indicated by the decrease in the magnetic field strength and, decrease in density and temperature and spectra expected for its entry into the solar wind. THEMIS A observed the IP shock at 16:49:12 UT and in about 1 min and 34 s later its magnetic field, density and temperature traces indicate that the bow shock moved inward past THEMIS A.

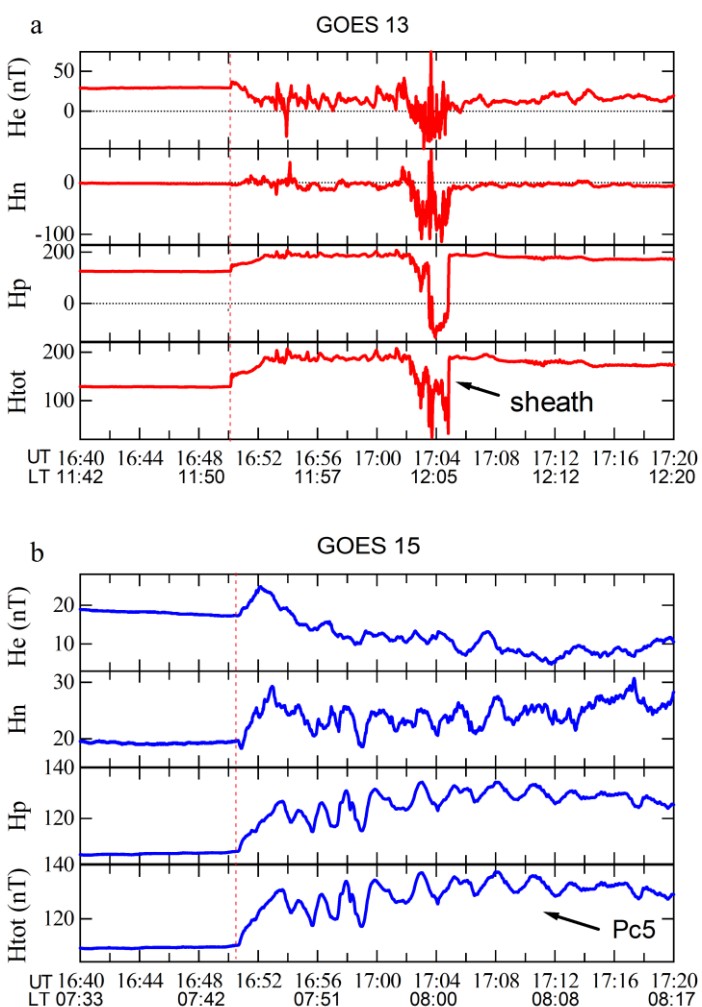

491

Figures 4 (a, b). GOES 13 (a) and GOES 15 (b) magnetic fields observations in PEN coordinate
from 16:40 UT to 17:20 UT on February 27, 2014. Hp is perpendicular to the satellite's orbital
plane, He pointing earthward parallel to the satellite-Earth center line, and Hn is perpendicular to
both Hp and He and pointing eastward.

496

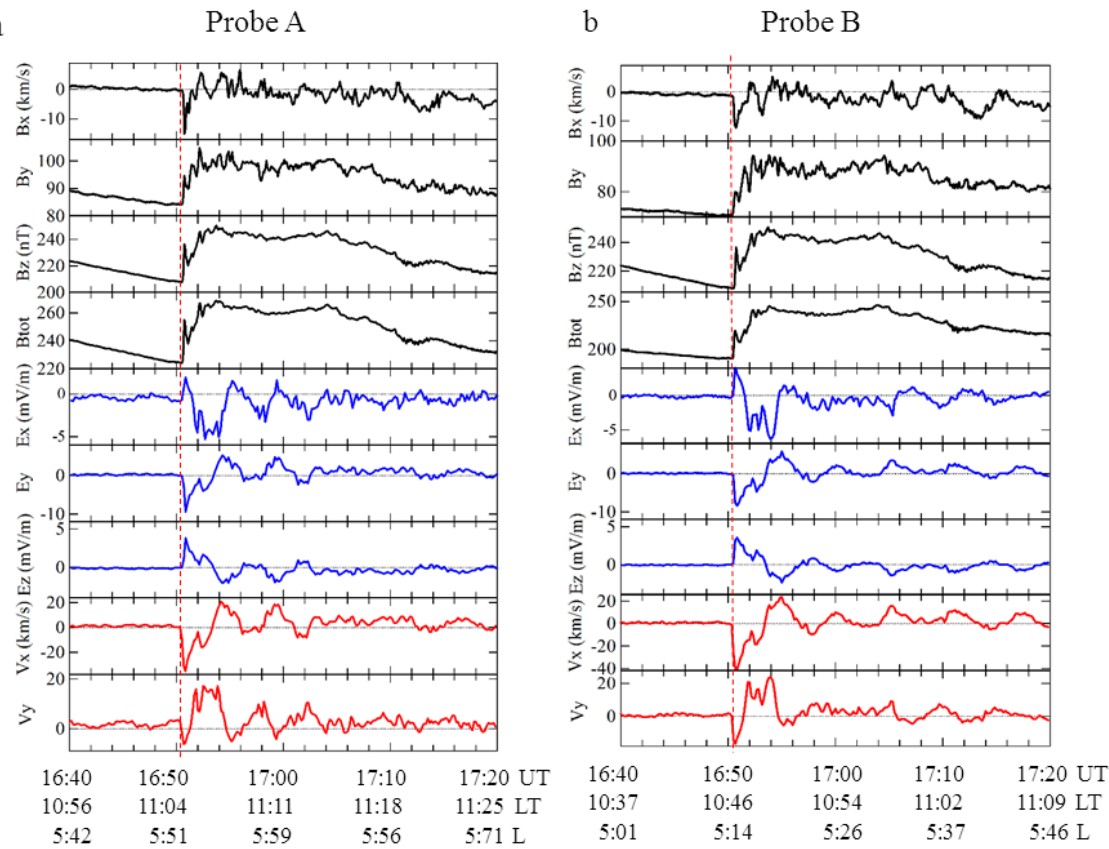

Figures 5 (a, b). Van Allen Probes A (a) and B (b) magnetic and spin-fit electric field observations and the Vx and Vy plasma flow velocities in GSE coordinates from 16:40 UT to 17:20 UT on February 27, 2014.

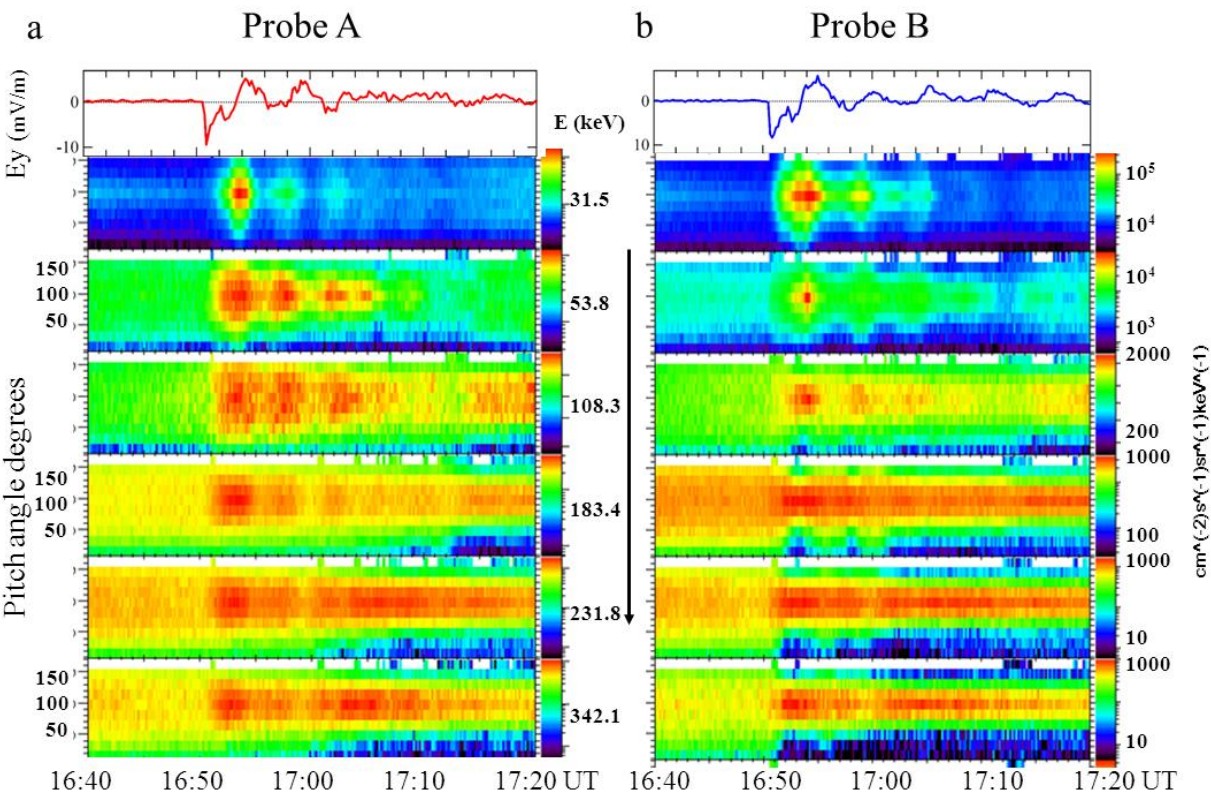


Figures 6 (a, b). Van Allen Probes A (a) and B (b) Ey component of the electric field and pitch
angle distributions of electron fluxes in the range of energies from 31.5 to 342 KeV, measured
by MagEIS instrument from 16:40 UT to 17:20 UT on February 27, 2014. The log fluxes are
color coded according to the color bar shown in the right panel.

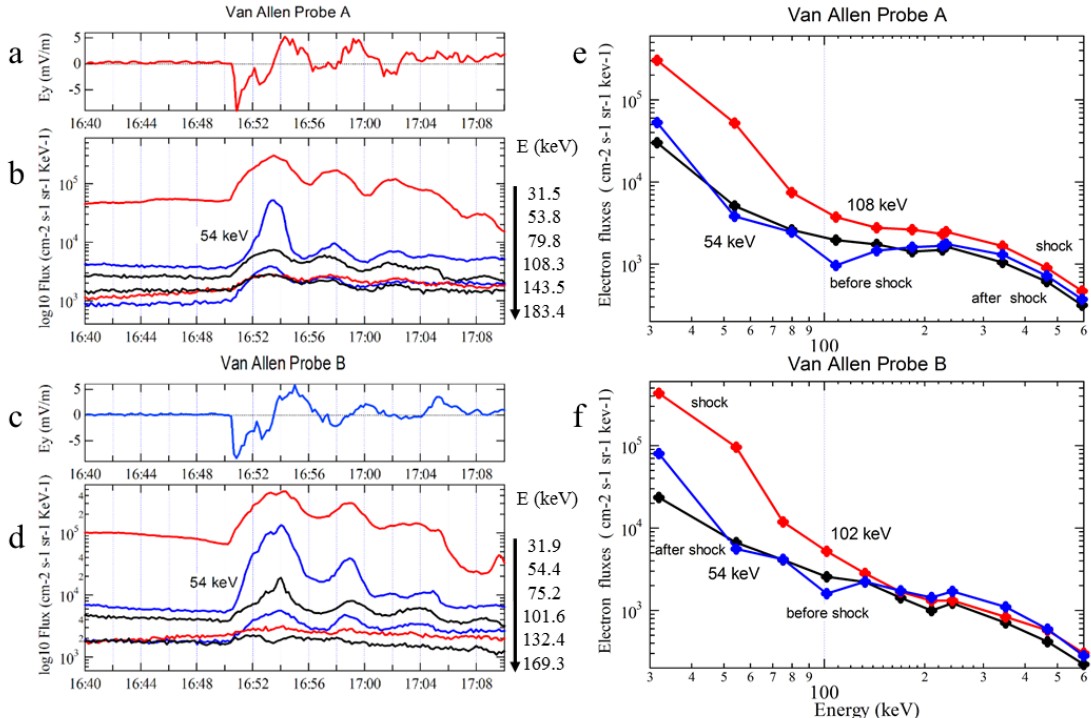


Figures 7. Response of the energetic particles to the transmitted IP shock. Panels a and c show
measurements of the Ey component of the electric field. Panels b and d show electron fluxes for
the energies ranging from 31.5 to 180 keV at Van Allen Probes A and B. Panels e and f show
energetic electron spectra observed at Probe A and B before the shock (at 16:48 UT), after the
first peak (at 16:53 UT) and 18 min after the shock (at 17:08 UT).








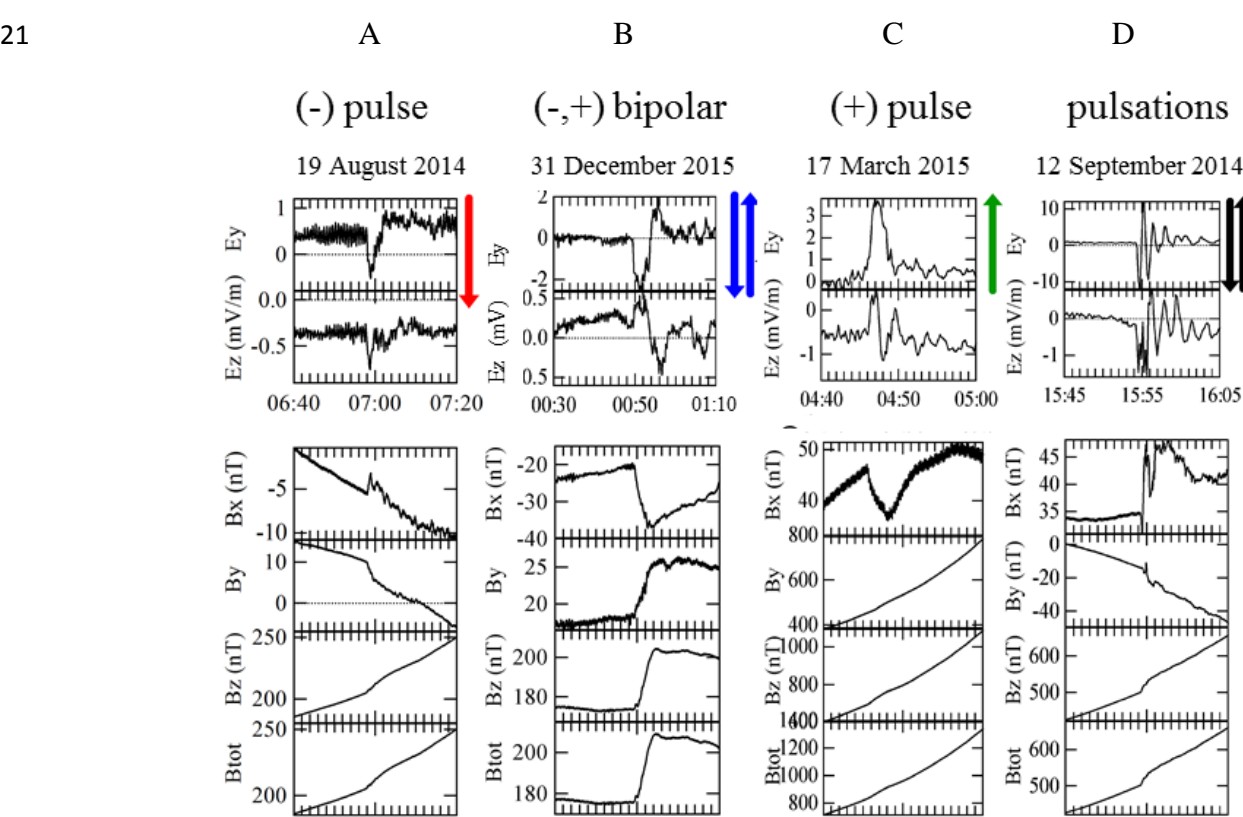


Figure 8.  Examples of observed Ey  initial variation, including a negative pulse (A), a negative-
positive waveform (B),  a positive pulse and  pulsations (upper panels)  and the corresponding
magnetic field response (bottom panels).

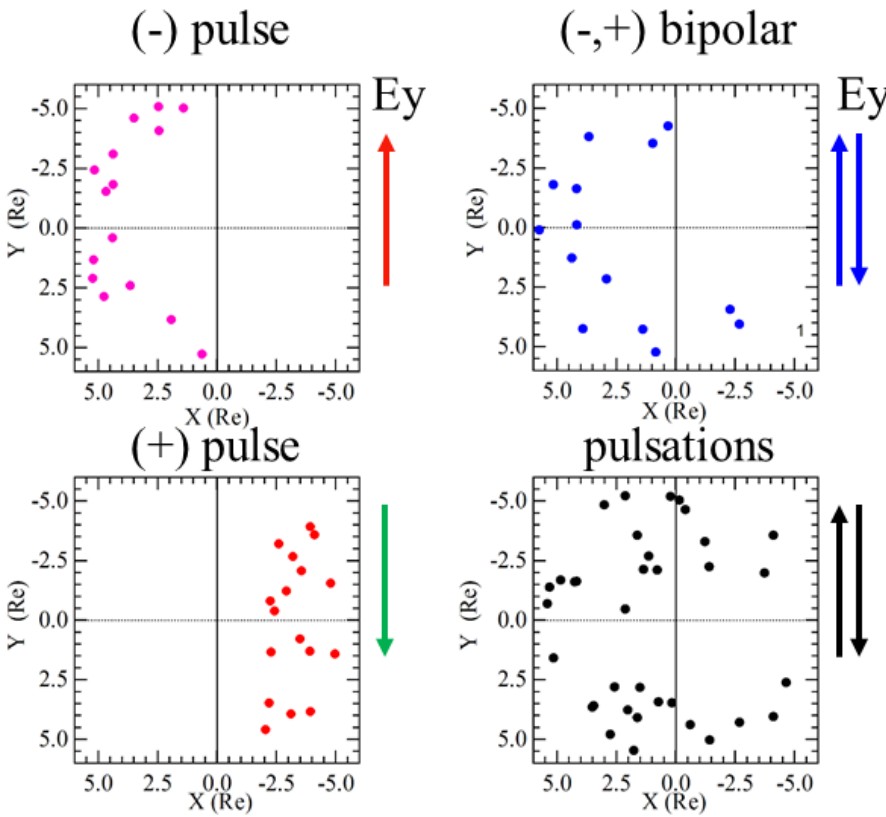


Figure 9.  GSM  locations  where event in each of the four groups were observed  in the X-Y
plane.

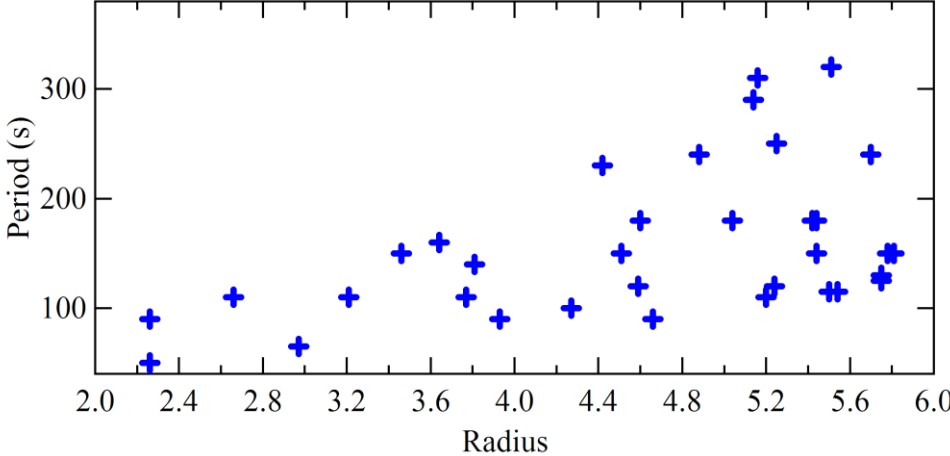


Figure 10. Periods of pulsations, initiated  by IP shocks  as a function of radius.

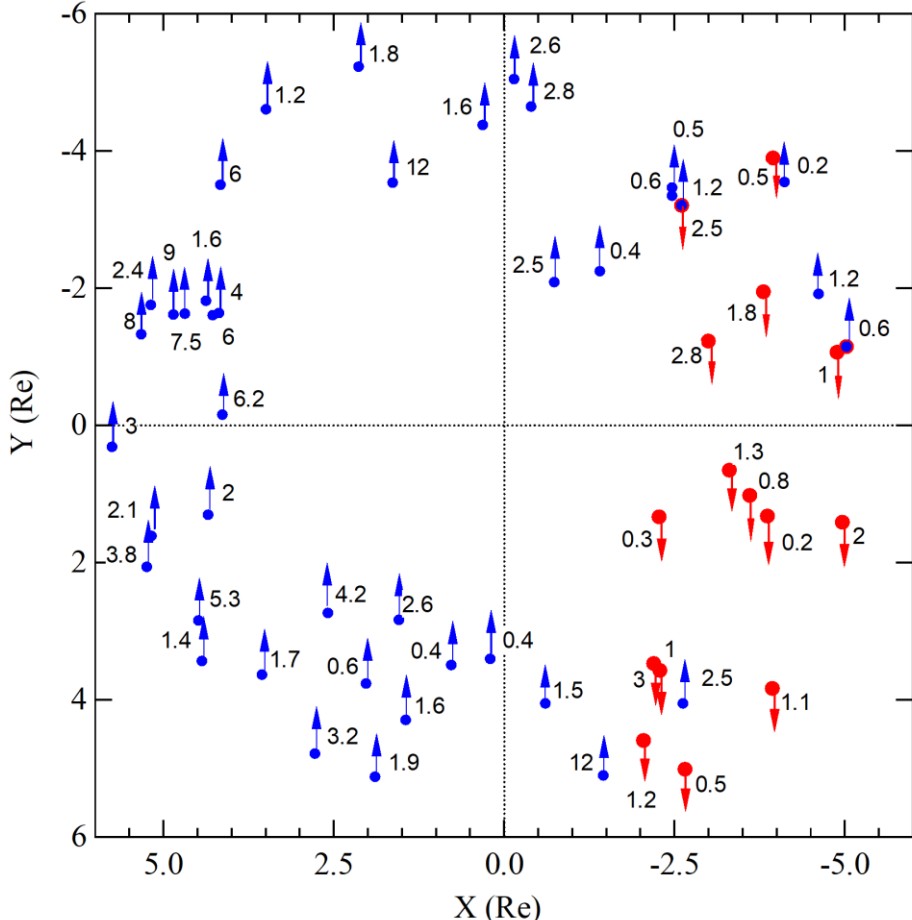


Figure 11. Amplitudes and direction of  initial Ey response to  IP shocks  in the X-Y GSM plane
(red   dawn-duskward and blue  duskward-dawn direction).

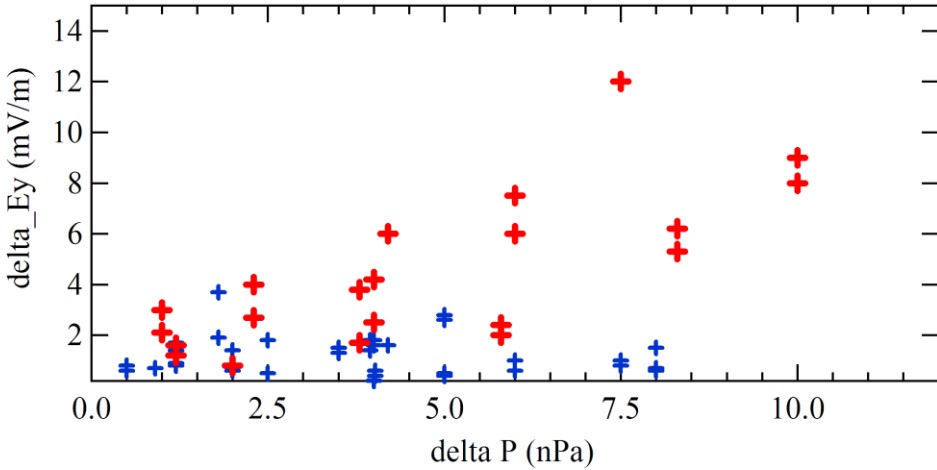


Figure 12. Amplitudes of initial  Ey variations  (blue and red points) caused by a shock    as
a function of intensity variations of  dynamic pressure   observed at  Wind for 60 events.

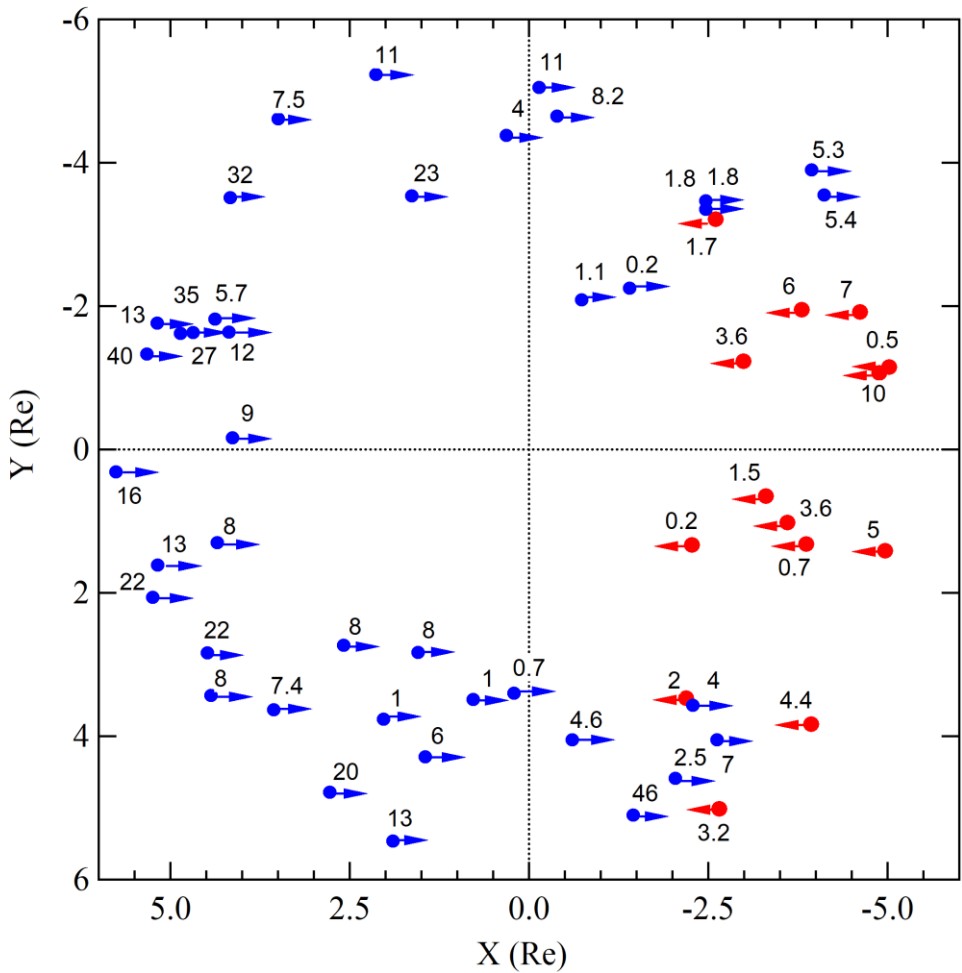


Figure 13. Amplitudes and direction of the plasma drift velocities Vx=ExB/B$^2$ observed by
Van Allen Probes A and B in response to interplanetary shocks (red - sunward and blue –
tailward direction).



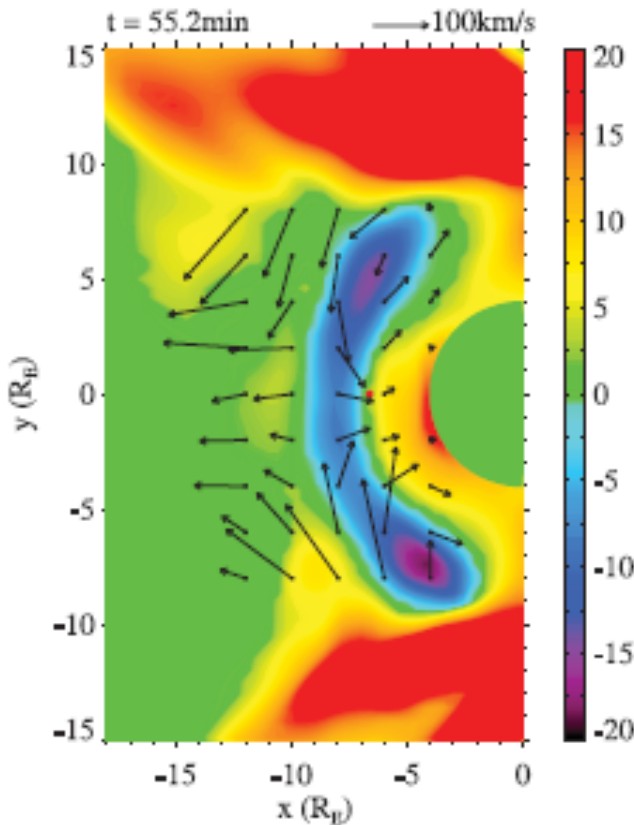


Figure 14. Results of nightside geosynchronous magnetic field response from the global MHD
code simulation of IP shock (Wang et al., 2010). The arrows represent velocity vectors on the
equatorial plane.



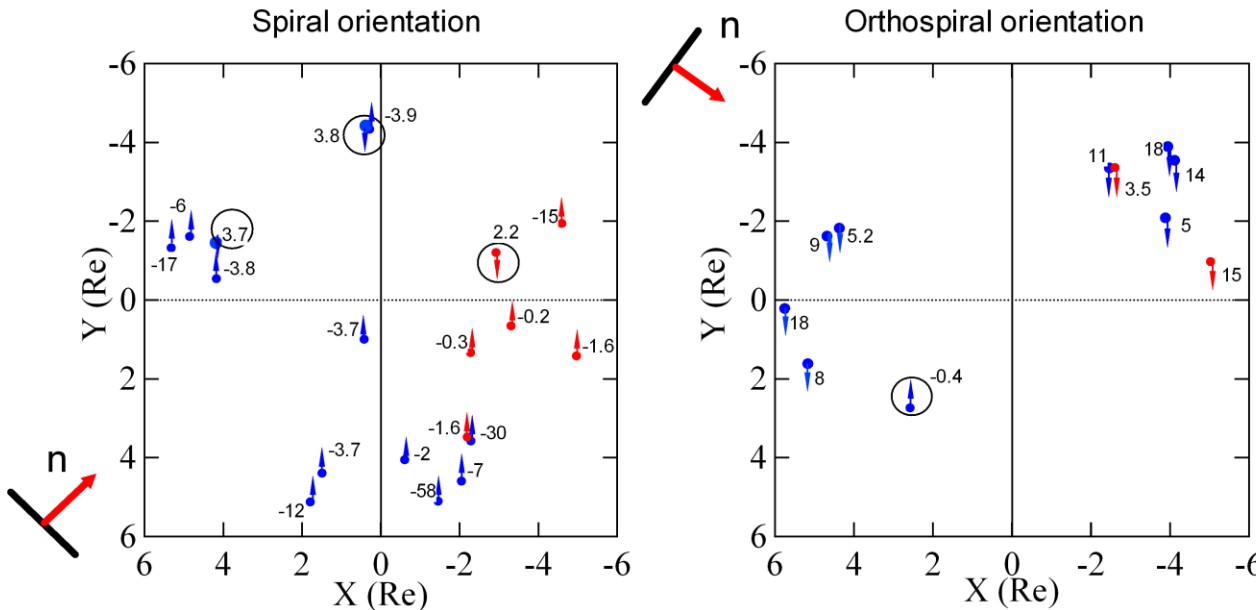

Figure 15. Amplitudes and direction of Vy plasma drift velocities observed by Van Allen
Probes A and B in response to interplanetary shocks for spiral and orthospiral orientations (red -
sunward and blue – tailward Vx directions).

## 6 Data availability.

Data used in the paper are available publicly at http://cdaweb.gsfc.nasa.gov/istp_public/ (Coor-
dinated Data Analysis Web). GOES data can be found at
http://satdat.ngdc.noaa.gov/sem/goes/data/new_full/. The electric field data were obtained from
sites http://www.space.umn.edu/rbspefw-data. The list of IP shocks used in this study was ob-
tained from site :http://ipshocks.fi.

**Team list:**
Galina Korotova, David Sibeck, Scott Thaller, John Wygant, Harlan Spence, Craig Kletzing,
Vassilis Angelopoulos, and Robert Redmon

**Author contributions:**
G. Korotova drafted and wrote the paper and all others commented it.
G. Korotova, D.Sibeck, S. Thaller and R. Redmon were the primary contributors to this work.
S. Thaller- programming, software development, consulting regarding use of Van Allen electric
field data.
J. Wygant -consulting regarding use of Van Allen electric field data.
H. Spence -consulting regarding use of Van Allen plsma data
C. Kletzing -consulting regarding use of Van Allen Magnetometer data
V. Angelopoulos -consulting regarding use of THEMIS data
The authors declare that they have no conflict of interest.
**Acknowledgements.** The Van Allen Probes mission is supported by NASA. NASA GSFC's
CDAWEB provided Wind and GOES observations, while SSCWEB provided Van Allen Probes
EPHEMERIS. The work by GIK at the University of Maryland was supported by grants  from
NASA NNX15AW86G S01 and NSF AGS-1207445. The work by the EFW team at the Univer-
sity of Minnesota was supported by APL contract to UMN 922613 under NASA contract to APL
NAS5-01072.

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
