# Peer review of "Multisatellite observations of the magnetosphere response to changes in the solar wind and inter planetary magnetic field"

_Annales Geophysicae, 2018_

## Referee Comment (RC1) · J. Foster (Referee) · 23 Apr 2018

This study addresses the characteristics of the important magnetospheric response tosolar wind shocks. Using multi-point in-situ observations within the magnetosphere the authors present case and statistical studies of the electromagnetic fields and plasma response during shock events. A detailed analysis of the strong February 27, 2014 shock event is presented. For the statistical investigation, 30 shock events of varied strength are investigated. The positions of the spacecraft making the in -situ observations differ event to event, enabling a discussion of the characteristic magnetospheric responses in the noon, midnight, dusk, and dawn sectors. The statistical

analyses concentrate on pulse propagation speed, direction, and strength (Ey) and the associated plasma drift velocity. Lastly, a brief comparison of the observed results with model calculations is presented.

The paper presents a good overview of previous studies relating to IP shock response within the magnetosphere and the descriptions of the data included in the study and the steps taken in reaching their results are presented clearly. Taken as a whole, the paper appears well written, accurate and believable. While the results of the studies presented are well documented, they do not seem particularly new. It is recommended that the authors add a bit more detail (see suggestions below e.g.) to show more clearly the significance of their results. It will be important for the authors to include a clear statement of what they feel are the most significant features of the study.

Suggestions:

For the fast mode propagation velocities, it would be good to describe the theoretical parametric dependence of the fast mode velocity (e.g. its dependence on radial distance). How well do the observed pulse velocities agree with theory for the Feb 2014 event (e.g. lines 310-312) and others?

lines 255-258: Two techniques for calculating the normal vector of the shock (n) are described. How well do these two techniques agree with each other?

In the Introduction (lines 89-100), the resonant acceleration of trapped particles is discussed briefly. This paper presents observations and calculations of the propagation speed of the shock-induced pulse, the strength and variation of Ey, and the associated plasma drift velocities Vx and Vy. A useful addition to the paper would be to present some detail on how those parameters have important effects on plasma acceleration in interactions involving the initial fast-mode pulse or with subsequent ULF oscillations.

For example, the studies of Wygant et al [1994] using CRRES data, Foster et al. [2015] using Van Allen Probes data, and others have shown that within the magnetosphere,

the tailward propagation of the strong shock-induced electric field impulse can result in the extremely fast acceleration of high energy, ultra-relativistic electrons deep within Earth's Van Allen radiation belts. The strong electric field associated with the shock-induced fast mode pulse is of about 1-min duration and accelerates radiation belt electrons for the length of time they are exposed to it.
* * *

---

## Short Comment (SC1) · 10 May 2018

In the case study presented in this paper, the interplanetary shock of interest has a significant GSM z-component to its normal vector. However, the paper appears really to only consider the in-ecliptic effects. For example, it concludes the point of first impact is near the nose of the magnetosphere/bow shock, when in a 3D consideration the point of impact might be some distance further northward and duskward? How does this affect the analysis/results? In the statistical study, do these effects show in the results also (e.g. with observations of N-S moving fronts)?

The comparison of the observational results with the simulation also seems a little

weak. The observational 'box' in Figure 9 has limited overlap with the simulation box in Figure 10, and there do not appear to be strong flows in the simulation in the overlap region. Is there no way to extend the observational coverage (e.g. Themis, Cluster) to include the region which has the strongest (and reversing) flows in the simulation?

———————————————————

---

## Short Comment (SC2) · 14 May 2018

Dear Dr. Foster, Thank you very much for your comments. Here is my response.

Foster:lines 255-258: Two techniques for calculating the normal vector of the shock (n) are described. How well do these two techniques agree with each other?

The Finnish data base gives the coordinates of the normal vector to shocks as calculated from the magnetic field data and velocities using the mixed mode method of Abraham-Shrauner and Yun [1976]. When there is data gap in the velocity components, the normal is calculated using magnetic field coplanarity [Colburn and Sonett,

1966]. Abraham-Shrauner [1972] suggested the "mixed mode method as an alternative to other methods when the accuracy of the magnetic field used in the calculations is uncertain. She noted that, for example, if the magnetic field is exactly normal or tangential to the shock front, magnetic coplanarity fails to give an expression for the shock normal. Our list of interplanetary shocks contains events for which the determination of the values of the magnetic field ahead and behind the shock was not complicated (no strong oscillations), so we always use magnetic field coplanarity to calculate the shock orientations. We found that the sense of our shock orientations (spiral or orthospiral) agrees well with the shock parameters in the Finnish database. For the fast mode propagation velocities, it would be good to describe the theoretical parametric dependence of the fast mode velocity (e.g. its dependence on radial distance). How well do the observed pulse velocities agree with theory for the Feb 2014 event (e.g. lines 310-312) and others?

Foster: For the fast mode propagation velocities, it would be good to describe the theoretical parametric dependence of the fast mode velocity (e.g. its dependence on radial distance). How well do the observed pulse velocities agree with theory for the Feb 2014 event (e.g. lines 310-312) and others?

To calculate VA we used the Carpenter and Anderson density profile obtained from a least squares linear fit to 25 ISEE dayside saturated plasmasphere profiles [J. Geophys. Res., 97, A2, 1097-1109, 1992]. Figure 1a shows their reference density profile given by ne =10(-0.3145L+3.9043) for L increments of 0.5. Figure 1b and Figure 1c show the values of the magnetic field obtained from a CCMC run for the Tsyganenko geomagnetic field model for the solar wind conditions on February 27, 2014 and the corresponding Alfven velocity, respectively. Then we set the plasmapause at L = 6 and took the density ne as 4 cm-3 beyond this distance to obtain the corresponding Alfven velocity presented in Figure 1d. The values for the Alfven velocity at the locations of Van Allen Probes A (L=5.1) and B (L=5.5) are about 284 km/s whereas at the GOES location it is 1240 km/s. Because the temperature is very low, the fast mode velocity is

the same as the Alfven velocity in the plasmasphere [Takahashi et al., J. Geophys. Res. 115, 2010] and is only ∼100 km/s greater in the outer magnetosphere. These model values for the fast mode velocities are in good agreement with the values obtained in our paper.

Foster: A useful addition to the paper would be to present some detail on how those parameters have important effects on plasma acceleration in interactions involving the initial fast-mode pulse or with subsequent ULF oscillations

Addition to the text (and to the Conclusions) : line 197: The most prominent increase in the electron intensities (by factors of 21 and 14) was observed for energies of 53.8 keV at Van Allen Probes B and A, respectively. Note that the impact of the shock on the relativistic electron populations observed by REPT was not significant and was characterized by a weak increase followed by a decrease. Line 226: Interaction with the initial fast mode pulse and subsequent ULF electrical field oscillations can have an important effect on particle acceleration. In considering the energization of electrons on February 27, 2014, an encounter with the observed electric field for a period of 240 s will transport the electrons earthward by ïAd'Re = 1.3 to 1.6 RE from their original position at L= 6.4 for Van Allen Probe A and at L = 7.1 for Van Allen Probe B. Conservation of the first adiabatic invariant implies that such particles will be energized by a factor of about 1.9 - 2.3 in only one cycle of the electric field pulsations. The studies of Wygant et al. [1994] using CRRES data and Foster et al., [2015] using Van Allen Probes data, and others have demonstrated that the tailward propagation of the strong shock-induced electric field impulse and subsequent ULF processes can result in the extremely fast acceleration of relativistic electron populations inside the plasmasphere.

[Figure]

**Fig. 1.** . Radial profiles: (a) reference density profile given by ne =10(-0.3145L+3.9043) for L increments of 0.5, (b) values of the magnetic field obtained from a CCMC run for the Tsyganenko geomagnetic fie

---

## Short Comment (SC3) · 30 May 2018

Dear Prof. Owen, Thank you for your helpful comments. With regards to the first point, a consideration of the 3D picture is indeed valuable, not least because it enables us to find an error in our calculation of the normal to the IP shock. The normal to this shock should point antisunward, dawnward, and northward. Consequently the shock should first strike the southern dusk bow shock and magnetopause. Although most of our spacecraft were located near the ecliptic plane, the Cluster spacecraft were fortuitously located at high southern postnoon latitudes. Consequently, as already noted in the paper, Cluster saw the event first (Table 1). THEMIS A and D then see the front nearly

[Figure]

simultaneously, suggesting it strikes a broad region of the magnetopause at once. We have therefore broadened the red impact eclipse on the magnetopause in Figure 2. The main purpose of our statistical study was to investigate whether the direction of the shock normal has any effect on the propagation of the shock induced magnetic and plasma disturbances in the X-Y plane. As professor Owen brought our attention to the importance of a 3D consideration for determination of the point of impact of the IP shock we calculated the Vz velocities for 15 events in our data base and obtained that for 12 events their directions are agreeable with z component of the normal to the IP shocks. As the results look encouraging, we plan to perform a further study of direction of plasma flow in response to IP shocks as well as to extend the observational coverage by using Cluster and THEMIS spacecraft in a new project. Sincerely, Galina Korotova.

[Figure]

**Fig. 1.** GSM locations of Cluster 1 and 2, THEMIS A, D, Van Allen Probes A and B and GOES 13 and 15 in the X-Y and Z-Y planes at $\sim$ 1650 UT on February 27, 2014. The meaning of the solid oval and thick

---

## Short Comment (SC4) · 9 Jun 2018

Figures 7 (a, b, c, d, e, f) present the response of energetic electrons to the IP shock. Panels (a, b, c, d) show a clear correlation between azimuthal electric field oscillations and energetic electron fluxes in the energy range from 31 to 183 keV that do not display obvious phase differences across the energies. After the shock arrival, the electron population increased, especially for the lower energies (panels b and d). In the electrical field of 14 mV/m and with a drift velocity of 35-40 km/s the flux of 54 keV electrons increased by factors of 21 and 14 at Van Allen Probes B and A, respectively, in less than a drift period (panels e and f). In the absence of parallel electrical fields the

[Figure]

energy variations of charged particles interacting with ULF waves in transverse magnetic field depends on two contributing parts: the magnetic field compression and the electric field acceleration [Southwood and Kivelson, 1982]. Zong et al. [2009] showed that the enhancement of the spectrum due to shock compression is rather small and suggested that the ULF electric field waves have a major contribution to the electron acceleration. Note that the impact of the shock on the relativistic electron populations observed on February 27, 2014 by REPT was not significant and was characterized by a weak increase followed by a decrease (not shown).
* * *
**Fig. 1.** Figures 7. Response of the energetic particles to the IP shock. Panels a and c show measurements of the azimuthal component of the electric field. Panels b and d show electron fluxes for the energies

---

## Referee Comment (RC2) · Anonymous Referee #2 · 13 Jun 2018

The purpose of the paper is to analyze the response of the magnetosphere to interplanetary shocks. First the authors present a case study for the event of February 27th 2014. They highlighted the main effects of this shock and presented interesting results on how this shock impacted the inner magnetosphere by comparing observations from Cluster, GOES and Van Allen Probes spacecraft. Then they performed a multicase study of 30 events observed by Van Allen Probes. Due to the different characteristics of the studied shocks and the observations at different locations, the authors have been able to present a global picture of the response in the magnetosphere to different type of shocks and provide interesting conclusions. However, I have some recommendations to the authors that I would like them to take into account before I

recommend the paper to publication.

The major one concerns the fact that it is quite difficult to find what is really new in this study. Of course the results are very interesting but they not provide perspectives or insights of what they could offer to inner magnetosphere scientists. First, the authors should highlight the main results in the abstract section. Then, at the end of the introduction section, it is not clear also what is the main purpose of this study and what it is new. Finally in the conclusion section, it is still not evident to find what is new compared to previous studies. The authors should try to improve this.

In this idea, I would like also to recommend the authors to analyze and discuss maybe a little bit more on the implications of their work regarding three directions:

- Using the muti-events analysis and their conclusions, is there a way to deduce from solar wind precursors, what will be the response of the magnetosphere : could we be able to estimate / anticipate the induced electric fields characteristics (directions, amplitudes, periods, . . .) that could be of interest regarding space weather (intensity, plasma heating, time lag. . .) ?

- Based on this analysis (bots the February 27th 2014 and the multicase study), some interesting perspectives / analysis could be made between the analyzed characteristics of the electrisc fields induced and the response of the radiation belts during these disturbed time especially regarding: dropouts at low energy induced by convection electric field (E < 100 keV) and radial transport trough typical radial diffusion for all energies?

- What is the impact of the plasmasphere in the dayside sector and in the nightside sector on induced electric fields at such times as the plasmasphere is no more circular (and conversely)?

As minor concerns :

- In the abstract section, I recommend the authors to mention the satellites used in their study.

- Line 122 : "data" mention twice
- Line 124 : "4-str" should be corrected to "4Pi-str"
- Line 162 : "the" used twice

---

## Referee Comment (RC3) · Anonymous Referee #3 · 25 Jun 2018

- Instead of GSE coordinates, a field-aligned coordinate system might be able to show in a more clear way the radial and azimuthal directions of propagation as well as the direction of the electric field with respect to the magnetosphere. For example, in Figure 6 the azimuthal components are discussed but Ey GSE is plotted.

- It is mentioned that "In the solar wind Cluster 2 observed the shock earlier then and Cluster 1, respectively, that is the shock moved dawnward". Was this supposed to mean "earlier than Cluster 1"? If yes, in Figure 2, C2 appears to be located dawnward of C1, so the shock should be moving duskward. Please clarify.

- It is written that "In the outer magnetosphere the propagation velocity for the disturbance was about 1348 km/s between Goes 13 and 15 but only about 390 km/s between Van Allen Probes B and A". These are greatly inconsistent, and this discrepancy is not discussed in the paper. To my understanding, this can only be reconciled if a different propagation direction is assumed for the red arrows of Figure 2, which might also require a reconsideration of the shock front propagation. A possible orientation could be an arrow that originates from the pre-noon region (e.g. 0900 LT) and points towards the Earth, which is different from the results of the paper. Please discuss.

- Please discuss in greater detail the methodology used in order to determine spiral and orthospiral orientations of the shock normal, and the expected errors in these estimates.

- The association of the four groups with ongoing processes could be further discussed. E.g., Pi pulsations and substorms are not mentioned at all in the paper.

---

## Author Response (AR1)

Dear Dr. Foster,

Thank you very much for your comments. Here is our response.

**lines 255-258: Two techniques for calculating the normal vector of the shock (n) are described. How well do these two techniques agree with each other?**

The Finnish data base gives the coordinates of the normal vector to shocks as calculated from the magnetic field data and velocities using the mixed mode method of Abraham-Shrauner and Yun [1976]. When there is data gap in the velocity components, the normal is calculated using magnetic field coplanarity [Colburn and Sonett, 1966]. Abraham-Shrauner [1972] suggested the "mixed mode method as an alternative to other methods when the accuracy of the magnetic field used in the calculations is uncertain". She noted that, for example, if the magnetic field is exactly normal or tangential to the shock front, magnetic coplanarity fails to give an expression for the shock normal. Our list of interplanetary shocks contains events for which the determination of the values of the magnetic field ahead and behind the shock was not very complicated (no strong oscillations), so we always use magnetic field coplanarity to calculate the shock orientations. We found that the sense of our shock orientations (spiral or orthospiral) agrees well with the shock parameters in the Finnish database.

**For the fast mode propagation velocities, it would be good to describe the theoretical parametric dependence of the fast mode velocity (e.g. its dependence on radial distance). How well do the observed pulse velocities agree with theory for the Feb 2014 event (e.g. lines 310-312) and others?**

The exact value of the fast mode wave speed in the magnetosphere depends on the direction of its propagation. For propagation perpendicular to B, the phase velocity $V_F$ is $(V_A^2 + C_S^2)$, where $V_A$ is Alfven velocity and $C_S$ is the sound velocity.

The Alfven velocity is given by $\mathbf{V_A} = \mathbf{B}/(\mu_0 \rho)^{-1/2}$. To calculate $V_A$ we used the Carpenter and Anderson density profile obtained from a least squares linear fit to 25 ISEE dayside saturated plasmasphere profiles [J. Geophys. Res., 97, A2, 1097-1109, 1992].

Figure 1a shows their reference density profile given by $n_e = 10^{(-0.3145L+3.9043)}$ for L increments of 0.5. Figure 1b and Figure 1c show the values of the magnetic field obtained from a CCMC run for the Tsyganenko geomagnetic field model for the solar wind conditions on February 27, 2014 and the corresponding Alfven velocity, respectively. Then we set the plasmapause at L = 6 and took the density $n_e$ as 4 cm$^{-3}$ beyond this distance to obtain the corresponding Alfven velocity presented in Figure 1d. The values for the Alfven velocity at the locations of Van Allen Probes A

(L=5.1) and B (L=5.5) are about 284 km/s whereas at the GOES location it is 1240 km/s.

Because the temperature is very low, the fast mode velocity is the same as the Alfven velocity in the plasmasphere [Takahashi et al., J. Geophys. Res. 115, 2010] and is only ~100 km/s greater in the outer magnetosphere.  These model values for the fast mode velocities are in good agreement with the values obtained in our paper.

[Figure]

**Figure 1.**  Radial profiles: (a) reference density profile given by $n_e = 10^{(-0.3145L+3.9043)}$ for L
increments of 0.5, (b) values of the magnetic field obtained from a CCMC run for the
Tsyganenko geomagnetic field model, corresponding Alfven velocities (c) without and (d) with
a plasmapause.

**In the Introduction (lines 89-100), the resonant acceleration of trapped particles is discussed briefly. This paper presents observations and calculations of the propagation speed of the shock-induced pulse, the strength and variation of Ey, and the associated plasma drift velocities Vx and Vy. A useful addition to the paper would be to present some detail on how those parameters have important effects on plasma acceleration in interactions involving the initial fast-mode pulse or with subsequent ULF oscillations. For example, the studies of Wygant et al [1994] using CRRES data, Foster et al. [2015] using Van Allen Probes data, and others have shown that within the magnetosphere, the tailward propagation of the strong shock-induced electric field impulse can result in the extremely fast acceleration of high energy, ultra-relativistic electrons deep within Earth's Van Allen radiation belts. The strong electric field associated with the shock- induced fast mode pulse is of about 1-min duration and accelerates radiation belt electrons for the length of time they are exposed to it.**

We  added new  Figures 7, 10, 11, 12, made some calculations,   rewrote the section of statistical study and  the conclusions to improve our paper.

Thank you again for your help.

Regards,

Galina Korotova.

Dear Referee2,

Thank you very much for your corrections and suggestions. We took your commemts seriously and it took some time to prepare our reply. Here is our reply.

**The major one concerns the fact that it is quite difficult to find what is really new in this study. Of course the results are very interesting but they not provide perspectives or insights of what they could offer to inner magnetosphere scientists. First, the authors should highlight the main results in the abstract section. Then, at the end of the introduction section, it is not clear also what is the main purpose of this study and what it is new. Finally in the conclusion section, it is still not evident to find what is new compared to previous studies. The authors should try to improve this.**

We have now highlighted the main results in the abstract and in the introduction discussing the purpose of the paper . We rewrote the conclusion of the paper.

**In this idea, I would like also to recommend the authors to analyze and discuss maybe a little bit more on the implications of their work regarding three directions:**

**Using the multi-events analysis and their conclusions, is there a way to deduce from solar wind precursors, what will be the response of the magnetosphere : could we be able to estimate / anticipate the induced electric fields characteristics (directions, amplitudes, periods, ...) that could be of interest regarding space weather (intensity, plasma heating, time lag...) ?**

We added a new Figure 10 and showed that the periods of the pulsations initiated by IP shocks increase with radius. We believe that most pulsations in the dayside magnetosphere at $L < 6$ are produced by field-line resonances.

Regarding space weather we added three additional Figures 11, 12 and 13 and a new paragraph in the statistical study section to describe the response of the magnetosphere to IP shocks. In particular we have a much more extensive discussion of electric field direction, amplitudes and period.

Electron perpendicular temperatures observed by HOPE were available for 30 events. 13 events showed an increase of temperature, 6 events showed a decrease of temperature and 11 events did not show any change. Proton perpendicular temperatures were available for 40 events. 24 events showed a decrease of T, 12 events showed an increase of T and 12 events did not show any change.  We did not find any consistent pattern for behavior of electron and proton
temperatures after  impact of IP shocks.

**Based on this analysis (both the February 27th 2014 and the multicase study), some**
**interesting perspectives / analysis could be made between the analyzed characteristics of**
**the electrisc fields induced and the response of the radiation belts during these disturbed**
**time especially regarding: dropouts at low energy induced by convection electric field (E <**
**100 keV) and radial transport trough typical radial diffusion for all energies?**

Here we are studying the immediate response to IP shocks. Studies of diffusion would require
determining ULF wave amplitudes, the extent of wave fields, and simulations which are beyond
the scope of this paper.

We added a paragraph to the paper:
Understanding and predicting such responses is important for reducing the risks associated with
space exploration. We found that 55 events showed an electron enhancement at energies of 32-
54 keV measured by MagEIS  at all local time  and three of them   were accompanied by
intensity  decreases at  higher  energies.  Five events   showed a decrease of the 32-54 keV
energy electrons observed  in the nightside magnetosphere.

**What is the impact of the plasmasphere in the dayside sector and in the nightside sector on**
**induced electric fields at such times as the plasmasphere is no more circular (and**
**conversely)?**

The figure  below presents  the magnitude of Vx flow velocities as a function of plasmaspheric
density obtained from  electric potential on the Van Allen Probes.  Consistent with expectations,
the velocities induced by IP shocks can attain greater values in regions of  low magnetosphere
densities and  are invariably small  for regions where  densities exceed 260 cm-3.

.

[Figure]

Amplitudes of shock induced Vx flow velocities as a function of plasmaspheric density obtained from electric potential on the Van Allen Probes.

We corrected minor errors.

Thank you again for your help.

G. Korotova

Dear Referee 3,

Thank you very much for your comments and suggestions.

Here is our reply.

**Instead of GSE coordinates, a field-aligned coordinate system might be able to show in a**
**more clear way the radial and azimuthal directions of propagation as well as the direction**
**of the electric field with respect to the magnetosphere. For example, in Figure 6 the**
**azimuthal components are discussed but Ey GSE is plotted.**

We used GSE coordinates because the magnetic field data were already available in these
coordinates. Our results suffice to show clearly a pattern for the radial and azimuthal directions
of propagation as well as the direction of the electric field in GSE coordinates. We replaced
"azimuthal" with Ey.

**It is mentioned that "In the solar wind Cluster 2 observed the shock earlier then and**
**Cluster 1, respectively, that is the shock moved dawnward". Was this supposed to mean**
**"earlier than Cluster 1"? If yes, in Figure 2, C2 appears to be located dawnward of C1, so**
**the shock should be moving duskward. Please clarify.**

Yes. We believe that the shock moved dawnward across the magnetosphere. The reason for this
is that we have used both Wind and Cluster observations to make numerous coplanarity
calculations of the normal to the IP shock for a wide variety of observed upstream and
downstream input parameters. After a useful online discussion with Prof. Owen we chose a very
typical normal that pointed in the (x, y, z) direction [-0.8, -0.4, -0.3]. From this, we conclude
that the shock propagated dawnward and antisunward and struck the duskside magnetopause
first, precisely consistent with the picture that we drew in the paper. We avoided discussing the
nz component as we get very mixed results for sense of this component depending on the input
parameters chosen.

The referee is correct. From timing considerations and the Cluster observations, it appears
that the shock should propagate duskward from C2 to C1. However, we are examining 3s time
resolution observations from Cluster, the two spacecraft are very close together, and the times
corresponding to the observations differ for the two spacecraft. The apparent lag from C2 to C1
is an artifact of the times when 3s observations are available. When we use C1 and C3, located
further apart, the lag time is most definitely consistent with dawnward and antisunward shock
propagation.

**It is written that "In the outer magnetosphere the propagation velocity for the disturbance was about 1348 km/s between Goes 13 and 15 but only about 390 km/s between Van Allen Probes B and A". These are greatly inconsistent, and this discrepancy is not discussed in the paper. To my understanding, this can only be reconciled if a different propagation direction is assumed for the red arrows of Figure 2, which might also re-quire a reconsideration of the shock front propagation. A possible orientation could be an arrow that originates from the pre-noon region (e.g. 0900 LT) and points towards the Earth, which is different from the results of the paper. Please discuss.**

Please, read our reply to Dr, Foster ( lines 25-47).

These model values for the fast mode velocities are in good agreement with the values obtained in our paper.

**Please discuss in greater detail the methodology used in order to determine spiral and orthospiral orientations of the shock normal, and the expected errors in these estimates.**

The Finnish IP shock data base gives the coordinates of the normal vector to shocks as calculated from the magnetic field data and velocities using the mixed mode method of Abraham-Shrauner and Yun [1976]. When there is data gap in the velocity components, the normal is calculated using magnetic field coplanarity [Colburn and Sonett, 1966].   Abraham-Shrauner [1972] suggested the "mixed mode method as an alternative to other methods when the accuracy of the magnetic field used in the calculations is uncertain".  She noted that, for example, if the magnetic field is exactly normal or tangential to the shock front, magnetic coplanarity fails to give an expression for the shock normal.  Our list of interplanetary shocks contains events for which the determination of the values of the magnetic field ahead and behind the shock was not very complicated (no strong oscillations).  We also removed some events with IMF radial orientation, so we can always use magnetic field coplanarity to calculate the shock orientations.  Though our calculations sometimes give different values for the normal vectors and depends on the choice of intervals chosen for the calculation, we repeatedly obtain the same sense of normal orientation for all the calculations and pairs of upstream and downstream values: dawnward or duskward for each of our calculations.  We also found that the sense of our shock orientations (spiral or orthospiral) agrees well with the shock parameters in the Finnish database.

**The association of the four groups with ongoing processes could be further discussed. E.g., Pi pulsations and substorms are not mentioned at all in the paper**

We added a paragraph and   a new Figure 11 in the paper.

The ULF electric field pulsations  of Pc and Pi types  produced by IP shocks are observed at all
local times and  in the range of  periods  from several tens of seconds to several minutes.  We
believe that  the magnetic field as well  the electric field pulsations   initiated by IP shocks are
generated by  a wide variety of mechanisms  including  plasma instabilities, transient
reconnection, pressure pulses, and   often correspond to field line resonances.   Their
characteristic   features   are determined to large extent by local time. In the dayside
magnetosphere typical pulsations are of the Pc5 type. Sometimes they last   for more than twenty
wave cycles without noticeable damping which could be explained by a continuous input of the
solar wind energy into the magnetosphere. In the nightside magnetosphere during substorms, the
generation of Pi2 pulsations   is more common. They exhibit an   irregular form, last 3-5 wave
cycles, and often exhibit damping. Figure 10 presents periods of the pulsations (measured for the
first wave cycle of oscillations) as a function of radius and shows that periods increase with
increasing radius.  A simple explanation for this behavior of pulsation frequencies with radial
distance can be given in terms of standing Alfvén waves along resonant field lines (Sugiura and
Wilson, 1964). The length of the field lines, the magnetic field strength, and the plasma density
distribution determine the Alfvén velocity, and the periods of the pulsations.   This plot indicates
that  most electric field pulsations of the Pc5 type in the dayside magnetosphere at $L < 6$  are
produced by  field line resonances.

We added some information on shocks and substorms.

Discontinuities in the solar wind plasma such as shocks have often been considered as possible
triggers for the release of energy stored within the magnetotail in the form of magnetospheric
substorms. Most previous studies of shocks leading to substorms have relied on ground
magnetometer observations. Recently it has been shown that the use of global  auroral images to
identify substorm onsets has some advantages over many other alternative substorm onset
signatures, such as  low-latitude Pi2 pulsations, auroral kilometric radiation (AKR), and
dispersionless particle injections and magnetic field dipolarization at geosynchronous
orbits  [e.g., Liou et al., 2003]. To identify substorms triggered by shocks in our study we
considered negative magnetic bays by examining the westward auroral electrojet  AL index at
the times when SSC were determined from low-latitude ground magnetograms.. As a
quantitative definition for the substorm bay does not exist we used the criteria of Liou et el. [
2003] that there should be a sharp decrease in AL of at least 100 nT occurring  within  a 20 min
window starting at the SSC. We found that  shocks triggered a substorm in the magnetosphere
in 17 of the 30 examined events. Further study  whether  these negative magnetic bays  are
unique identifiers of substorms is beyond the scope of the paper**.**

Thank you again for your help with improving the paper.

Regards,

Galina Korotova.